# Post-Pemphigus Acanthomata Presenting as an Isolated, Hyperkeratotic Plaque

Rachita Pandya [1], Yanjia Zhou [2], Mansee Desai [2], Nancy Anderson [2] and Ashley Elsensohn [2,*]

1   California University of Science and Medicine, Colton, CA 92324, USA
2   Department of Dermatology, Loma Linda University, Loma Linda, CA 92354, USA
*   Correspondence: aelsensohn@llu.edu; Tel.: +1-909-558-2167

**Abstract:** Post-pemphigus acanthomas have been rarely discussed in the literature. A prior case series identified 47 cases of pemphigus vulgaris and 5 cases of pemphigus foliaceus, out of which 13 developed acanthomata as a part of the healing process. Additionally, a case report by Ohashi et al. reported similar recalcitrant lesions on the trunk of a patient with pemphigus foliaceus being treated with prednisolone, IVIG, plasma exchange, and cyclosporine. Some view post-pemphigus acanthomas as variants of hypertrophic pemphigus vulgaris, being difficult to diagnose when they present as only single lesions, with a clinical differential of an inflamed seborrheic keratosis or squamous cell carcinoma. Here, we present a case of a 52-year-old female with a history of pemphigus vulgaris and four months of only topical therapy (fluocinonide 0.05%) who presented with a painful, hyperkeratotic plaque on the right mid-back that was found to be a post-pemphigus acanthoma.

**Keywords:** pemphigus vulgaris; hyperkeratosis; acanthosis; acantholysis

## 1. Introduction

Pemphigus is an umbrella term describing a group of autoimmune vesicobullous diseases of the skin and mucosa [1]. While there exist several clinical variants of pemphigus, they all share similar histopathologic features [1]. All variants are characterized by disruption of normal cell adhesion (acantholysis) and circulating antibodies directed towards antigens on keratinocytes [1]. Once a pemphigus blister ruptures, it generally leaves behind erosions that do not spontaneously resolve but rather continue to grow through confluence [1]. With treatment, most lesions progress to dyspigmentation and then eventually resolve [2]. However, there have been reported cases of verrucous lesions developing at the sites of ruptured bullae and at sites of previously healed blisters in patients with pemphigus.

Post-pemphigus acanthomata is a rare but clinically significant lesion that can occur in individuals with pemphigus [2]. It typically presents as a verrucous lesion at a site of a previous blister, and can occur in individuals with clinical remission of pemphigus vulgaris or pemphigus foliaceus [2]. Histopathology of the lesion shows acantholysis resulting in intra-epidermal clefts, hyperkeratosis, papillomatosis, and acanthosis [2]. Here, we present a case of a patient with post-pemphigus acanthomata.

## 2. Case Synopsis

A 52-year-old female with a history of pemphigus vulgaris presented with a painful, hyperkeratotic plaque on the right mid-back. She denied a history of non-melanoma skin cancer. Prior treatments for her pemphigus vulgaris included prednisone, mycophenolate mofetil, rituximab, and intravenous immune globulin (IVIG). At the time of presentation, the patient had no active blisters or erosions and had been off of prednisone, mycophenolate mofetil, and IVIG for four months. She was using only topical fluocinonide 0.05% ointment five times a week, twice daily. On physical examination, an isolated 1.8 cm × 1.1 cm hyperkeratotic lesion was observed with some surrounding erythema (Figure 1). Histopathologic

examination of a shave biopsy demonstrated hyperkeratosis, acanthosis, focal hypergranulosis, and acantholysis extending down a hair follicle versus broad, elongated rete (Figure 2). The histologic differential diagnosis included an acantholytic acanthoma, but given the patient's longstanding history of pemphigus vulgaris and possible extension of acantholysis down the follicular epithelium, a diagnosis of post-pemphigus acanthomata was favored. Immunofluorescence was not performed given the high clinical suspicion in the context of the patient's prior diagnosis of pemphigus. Of note, the patient's anti-desmoglein 1 and anti-desmoglein 3 antibody titers were 3.28 and 198.98, respectively, at the time of diagnosis. At the time of biopsy, the anti-desmoglein 1 and anti-desmoglein 3 antibody titers were 3.14 and 180.60, respectively. The patient was started on doxycycline 100 mg twice daily, mycophenolate mofetil titrated to 1 gram twice daily, and continued on topical fluocinonide 0.05% ointment twice daily as needed for any new lesions. The plaque healed after biopsy and the patient did not develop any new areas of pemphigus with the aforementioned regimen. She has remained in remission for the past 8 months.

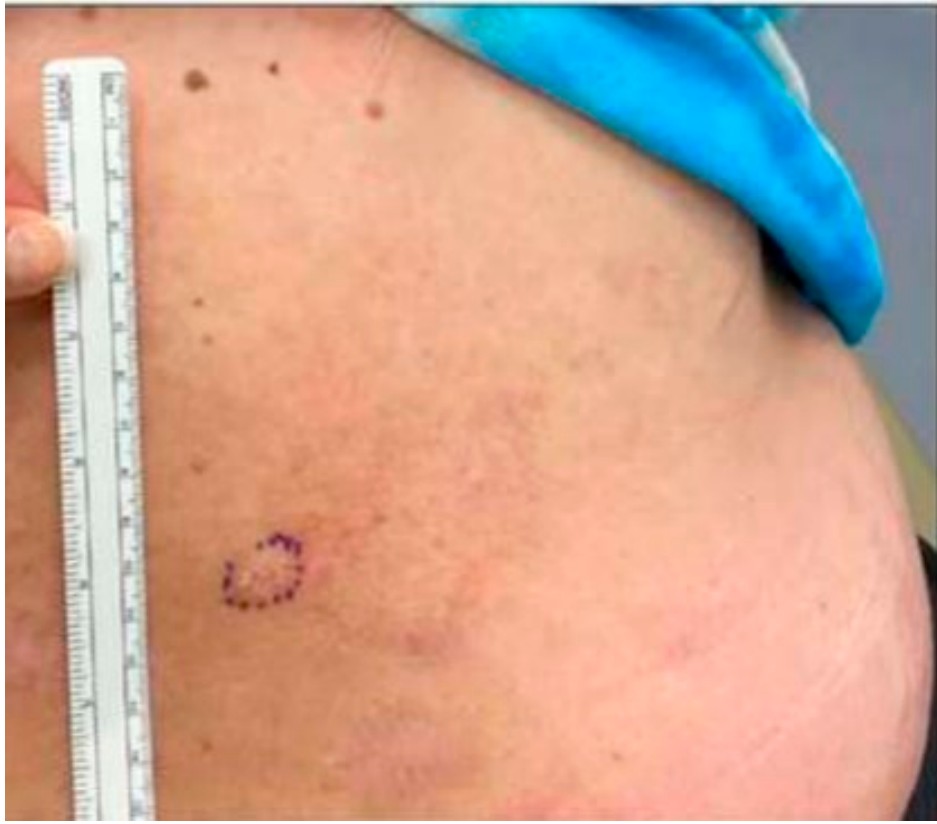

**Figure 1.** Clinical appearance of 1.8 cm × 1.1 cm hyperkeratotic plaque on the right mid-back.

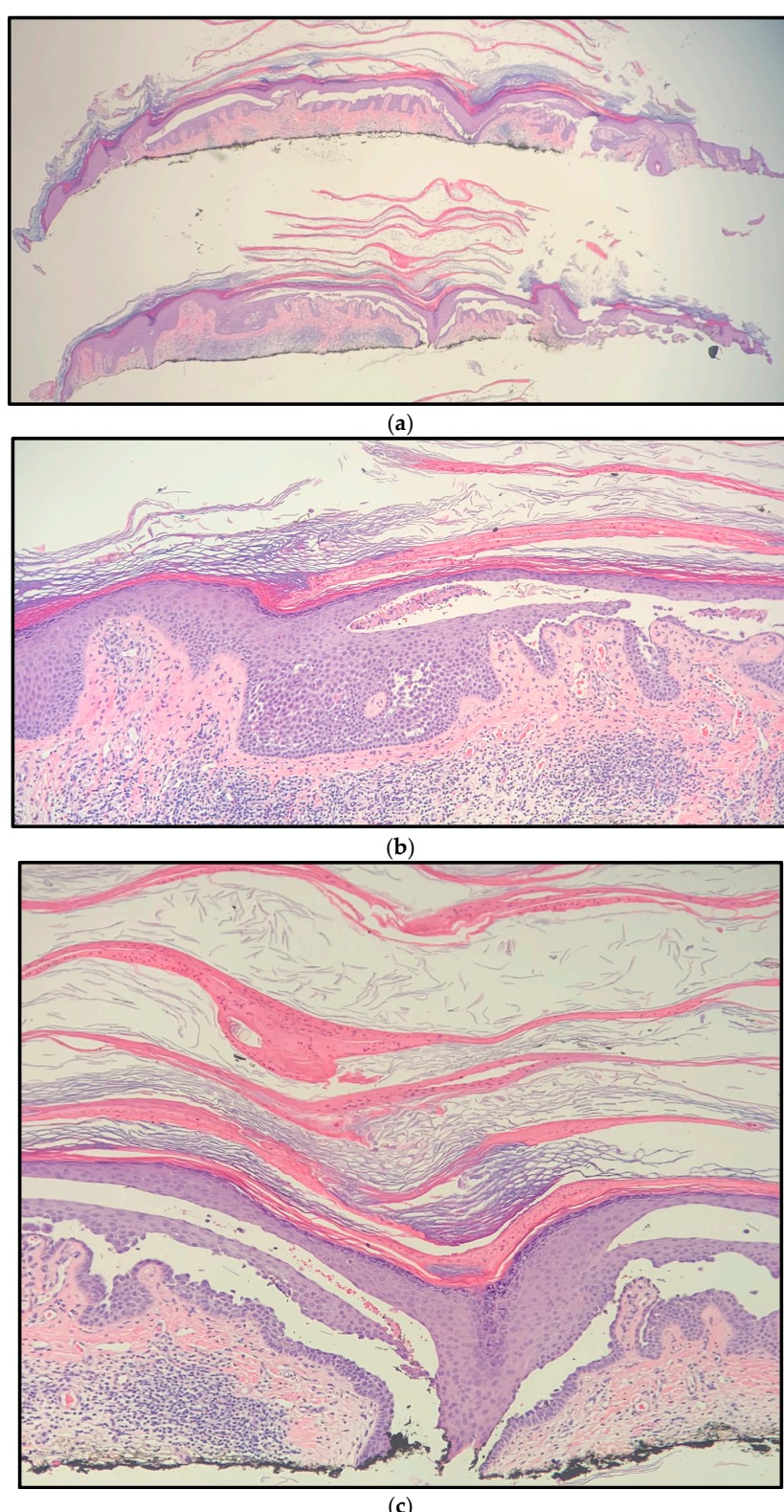

**Figure 2.** Right mid-back shave biopsy histology (hematoxylin and eosin) showing hyperkeratosis, acanthosis, focal hypergranulosis, and acantholysis extending down a hair follicle versus elongated, broad rete at magnification: (**a**) ×10; (**b**) ×20; (**c**) ×40.

### 3. Discussion

Post-pemphigus acanthomata has rarely been discussed in the peer-reviewed literature. To date, there is a limited body of information regarding this clinical entity. A case series by Yesudian et al. discussed 52 pemphigus patients (47 pemphigus vulgaris and 5 pemphigus foliaceous), 13 of whom developed acanthomas during a period of clinical remission [2]. The acanthomas presented as verrucous lesions at sites of prior blisters, and mainly occurred on the trunk and extremities [2]. Each lesion was hyperpigmented and had a "stuck on" appearance [2]. Histopathologically, the lesions showed intraepidermal clefts associated with acanthosis, hyperkeratosis, and papillomatosis [2]. On direct immunofluorescence, the acanthomas were found to have the characteristic pattern of pemphigus [2].

Another case report by Ohashi et al. identified treatment-resistant keratotic verrucous plaques that developed on the chest and abdomen of a pemphigus foliaceous patient, starting in year 5 of an 11-year treatment course [3]. The patient was being treated with prednisolone, IVIG, plasma exchange, and cyclosporine [3]. These lesions gradually developed and persisted for over six years [3]. Histological analysis depicted epidermal hyperplasia with regular papillomatosis and acantholysis in the granular layers [3].

Kucukoglu et al. also reported scalp lesions in patients with pemphigus [4]. In two pemphigus vulgaris patients with mucocutaneous involvement, treatment-resistant, vegetative scalp lesions associated with cicatricial alopecia developed at two years and four years following diagnosis, respectively [4]. Furthermore, these lesions persisted despite disease remission [4]. Histopathologic examination revealed an acanthotic epidermis and suprabasal acantholytic cleavage in the hair follicles [4].

In addition to presenting cases of post-pemphigus acanthomata, each one of these reports also importantly highlighted that acanthomata can develop anytime during the course of remission, after remission, or during relapse of pemphigus [2–4]. In addition to post-remphigus acanthomata, acanthosis nigricans-like hyperkeratotic lesions have also been documented during or after re-epithelialization in pemphigus [3]. Reported cases of these lesions were transient and spontaneously regressed thereafter.

With only a handful published outcomes of acanthomata in patients with a history of pemphigus, some argue that post-pemphigus acanthomas are a hypertrophic variant of pemphigus vulgaris, as may be demonstrated in our case with subtle erythema around the hyperkeratosis, in addition to an elevated anti-desmoglein 3 at the time of presentation (with a similar value at the time of diagnosis) [2–4]. Active pemphigus vulgaris is classically characterized by multiple mucocutaneous blisters that may even appear hyperkeratotic [5]. Acantholysis resulting in a suprabasilar split would be present on histology, as in this case. This can be difficult to identify however, given the solitary nature of post-pemphigus acanthomata in our case, in a patient with no recent flares.

Post-pemphigus acanthomata must also be distinguished from squamous cell carcinoma and inflamed seborrheic keratosis clinically and Hailey–Hailey disease and acantholytic acanthoma histologically. Squamous cell carcinoma can present as a solitary, hyperkeratotic plaque or nodule [6]. However, histopathology would show full thickness epidermal keratinocyte dysplasia, which is not seen in a post-pemphigus acanthomata. Hailey–Hailey disease, also known as benign chronic familial pemphigus, presents with red scaly areas, erosions, and vesicles or bullae that can result in hyperpigmented patches [7]. While this presents histologically similar to post-pemphigus acanthomata with acantholysis, Hailey–Hailey disease does not have discrete lesions clinically [8]. Moreover, the acantholysis in Hailey–Hailey does not extend down hair follicles, whereas post-pemphigus acanthomata does demonstrate this feature. Acantholytic acanthomas can present as solitary lesions and show no tendency for spontaneous regression, like the present case, but the acantholysis in acantholytic acanthomas does not extend to the hair follicles (which our case was likely to have), helping distinguish it from post-pemphigus acanthomata [2].

Prior studies have also shown that a negative direct immunofluorescence (DIF) finding is a good indicator of remission in pemphigus vulgaris [9]. However, Yesudian et al. carried out immunofluorescence on two of the identified acanthomata and found intercellular

fluorescence [2]. Consequently, evidence showing positive DIFs in post-pemphigus acanthomata suggests that these lesions may be a herald sign of active disease necessitating continued treatment and close follow-up, as they represent a hyperkeratotic variant of pemphigus vulgaris.

Management of post-pemphigus acanthomata typically requires systemic treatment. The current treatment of choice for pemphigus vulgaris includes options such as oral corticosteroids, steroid sparing immunosuppressive medications, rituximab, and IVIG [4]. Because post-pemphigus acanthomata mimics the histopathologic and immunofluorescence activity of the underlying pemphigus, there is potential for relapse [2]. Clinicians must continue pemphigus therapy to maintain remission.

### 4. Conclusions

We present a case of post-pemphigus acanthomata or hypertrophic pemphigus vulgaris, in a middle-aged female with a history of pemphigus vulgaris but no recent disease flares. In contrast to widespread active disease, post-pemphigus acanthomata may present as solitary lesions that can develop any time during the course of remission, after remission, or during relapse of pemphigus. Because these acanthomas show histopathological and immunofluorescent activity, they are clinically significant entities that require close follow-up and maintenance therapy.

**Author Contributions:** Writing—original draft preparation, R.P. and A.E.; writing—review and editing, R.P., Y.Z., M.D., N.A. and A.E.; project administration, data curation, A.E. All authors have read and agreed to the published version of the manuscript.

**Funding:** This research received no external funding.

**Institutional Review Board Statement:** Not applicable.

**Informed Consent Statement:** Written informed consent has been obtained from the patient(s) to publish this paper.

**Data Availability Statement:** Not applicable.

**Conflicts of Interest:** The authors declare no conflict of interest.

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
