# Peer review of "Post-Pemphigus Acanthomata Presenting as an Isolated, Hyperkeratotic Plaque"

_dermatopathology, doi:10.3390/dermatopathology10010012_

Round 1

Reviewer 1 Report

The authors report a rare case of post-pemphigus acanthoma, but there are several problems that need to be improved.

1) The Results section is completely lacking.

2) The clinical image in Figure 1 is out of focus.

3) Figure 1 shows multiple red erythema around the hyperkeratotic nodule. Is this not a sign of relapse? What were the anti-desmoglein 1 and desmoglein 3 antibody titers?

4) Spell out IVIG.

5) Cellcept is a trade name. Use the generic name.

6) Insert the citation after vulgaris in line 84.

Reviewer 2 Report

1. A diagnosis of post pemphigus acanthomata, although, is the plausible diagnosis considering the prior history of pemphigus, however, DIF study would have been confirmatory. This lesions might be an early sign of disease reactivity, but in the absence of DIF study and without subsequent course of the illness, a possibility of acantholytic acanthoma cannot be entirely excluded. More so for the reason that in the prior reports, such lesions are multiple and occurred during the course of disease remission or during relapse, even in those during developing after clinical remission, DIF was positive indicative of disease activity. Although, to my knowledge, there has been no published report of acantholytic acanthoma occurring in pemphigus, but there has been rare published reports of Darier disease and Grover disease with pemphigus. In one of the previous series on pemphigus acanthomata, one of the main distinguishing feature to differentiate from pemphigus acanthomata from acantholytic acanthoma is solitary lesion in later, therefore, in the presented case with solitary lesions and no other clinical evidence of activity or immunofluorescence, a diagnosis is questionable.

2. Apart from two case reports cited by authors, there is a related report published recently : “The above article is Kucukoglu R, Atci T, Babuna-Kobaner G, Buyukbabani N. Recalcitrant vegetative and nodular scalp lesions on the vertex in pemphigus patients: an immunocompromised district? An Bras Dermatol. 2022 Oct 29:S0365-0596(22)00226-4”.  Additionally, there are similar reports of hyperkeratotic lesions developing in the re-epithelizing lesions of pemphigus wherein term acanthosis nigricans like lesions is being used instead of the term acanthomata or verrucous, but these cases also represent the similar phenomenon.

3. The clinical image is not very clear and fails to depict the hyperkeratotic nature.

Round 2

Reviewer 1 Report

 Although it is unfortunate that the authors cannot provide an improved version of Figure 1 with higher resolution or DIF in the revised manuscript, my questions and comments have been adequately answered.

Author Response

Thank you for your time and review of our manuscript. Your suggestions were greatly appreciated. 

Reviewer 2 Report

1. Authors may consider to briefly mention about the acanthosis nigricans like lesion. It is not essential to add this, but it would make the report more comprehensive.  It can be incorporated at the end of first paragraph under discussion, following the details of previous reports of post pemphigus acanthomata. Authors can just add that apart from post pemphigus acanthomata, acanthosis nigricans like hyperkeratotic lesions have been documented during or after re-epithelization in pemphigus. For details on these lesions, authors can refer to, as well as cite reference 2 i.e. Ohashi T, Ohtsuka M, Kikuchi N, Yamamoto T. Verrucous variant of pemphigus foliaceus. Clin Exp Dermatol. 2020 141 Jul;45(5):584-585. doi: 10.1111/ced.14141. Epub 2019 Dec 19. PMID: 31725926. This article cites a couple of previously published acanthosis nigricans like lesions in pemphigus. I am not sure if the cases cited in the above article are the only cases of such lesions published in literature or there are more such cases. Authors may like to check this aspect.

 2. It would be valuable if authors clearly write that acanthomata can develop during the course of remission or after remission or during relapse (as evident from three previous reports on this entity- references 1-3).

3. Since acanthomata may herald the relapse in pemphigus, it is important to provide further follow-up of the patient.  That is whether patient developed more acanthomata or vesicular lesions following this single acanthomata and if this single lesion resolved on its own or following treatment if so what treatment.

4. I agree with the authors that acantholysis involving hair follicle is a helpful feature to distinguish pemphigus from other acantholytic disorders such as HHD. However, the image provided does not depict this feature. There is a small portion going down but it appears to be broad endophytic rete ridge and is not convincing to me for a hair follicle. Authors must try to provide an image with an unequivocal follicular structure to substantiate this finding.

5. Authors have included acantholytic acanthoma as differential diagnosis for the present case under case synopsis. However, authors have omitted this important differential under discussion where authors have discussed other differentials diagnosis of HHD, SK, SCC. Yesudian et al give two pointers for it; first that acantholytic acanthoma is a single lesions, second that it shows no tendency towards spontaneous regression. The lesions being single in the case presented, it becomes all the more reason for authors to give arguments against the diagnosis of acantholytic acanthoma under discussion section.

Author Response

Reviewer 2

  1. Authors may consider to briefly mention about the acanthosis nigricans like lesion. It is not essential to add this, but it would make the report more comprehensive. It can be incorporated at the end of first paragraph under discussion, following the details of previous reports of post pemphigus acanthomata. Authors can just add that apart from post pemphigus acanthomata, acanthosis nigricans like hyperkeratotic lesions have been documented during or after re-epithelization in pemphigus. For details on these lesions, authors can refer to, as well as cite reference 2 i.e. Ohashi T, Ohtsuka M, Kikuchi N, Yamamoto T. Verrucous variant of pemphigus foliaceus. Clin Exp Dermatol. 2020 141 Jul;45(5):584-585. doi: 10.1111/ced.14141. Epub 2019 Dec 19. PMID: 31725926. This article cites a couple of previously published acanthosis nigricans like lesions in pemphigus. I am not sure if the cases cited in the above article are the only cases of such lesions published in literature or there are more such cases. Authors may like to check this aspect.

Thank you for your time and constructive feedback. Your continued review and suggestions are much appreciated and have greatly improved the paper. We have now included the ancanthosis nigricans-like post pemphigus lesions in the first paragraph of the Discussion section.

  1. It would be valuable if authors clearly write that acanthomata can develop during the course of remission or after remission or during relapse (as evident from three previous reports on this entity- references 1-3).

This has now been included in the Discussion and Conclusion sections.

  1. Since acanthomata may herald the relapse in pemphigus, it is important to provide further follow-up of the patient. That is whether patient developed more acanthomata or vesicular lesions following this single acanthomata and if this single lesion resolved on its own or following treatment if so what treatment.

We have now included the following information about the patient’s disease course after biopsy: The patient was started on doxycycline 100mg twice daily, mycophenolate mofetil titrated to 1 gram twice daily, and continued on topical topical fluocinonide 0.05% ointment twice daily as needed for any new lesions. The plaque healed after biopsy and the patient did not develop any new areas of pemphigus with the aforementioned regimen. She has remained in remission for the past 8 months.

  1. I agree with the authors that acantholysis involving hair follicle is a helpful feature to distinguish pemphigus from other acantholytic disorders such as HHD. However, the image provided does not depict this feature. There is a small portion going down but it appears to be broad endophytic rete ridge and is not convincing to me for a hair follicle. Authors must try to provide an image with an unequivocal follicular structure to substantiate this finding.

Unfortunately, this is the best section that we have to demonstrate this feature. We have pointed out this limitation within the paper. 

  1. Authors have included acantholytic acanthoma as differential diagnosis for the present case under case synopsis. However, authors have omitted this important differential under discussion where authors have discussed other differentials diagnosis of HHD, SK, SCC. Yesudian et al give two pointers for it; first that acantholytic acanthoma is a single lesions, second that it shows no tendency towards spontaneous regression. The lesions being single in the case presented, it becomes all the more reason for authors to give arguments against the diagnosis of acantholytic acanthoma under discussion section.

These points have been included in the manuscript Discussion section. Thank you again for your time and review of our paper.
